# Immune Response to Natural and Experimental Infection of *Panulirus argus* Virus 1 (PaV1) in Juveniles of Caribbean Spiny Lobster

**DOI:** 10.3390/ani12151951

**Published:** 2022-08-01

**Authors:** Cristina Pascual, Rossanna Rodríguez-Canul, Juan Pablo Huchin-Mian, Maite Mascaró, Patricia Briones-Fourzán, Enrique Lozano-Álvarez, Ariadna Sánchez, Karla Escalante

**Affiliations:** 1Unidad Multidisciplinaria de Docencia e Investigación, Facultad de Ciencias, Universidad Nacional Autónoma de México, Puerto de Abrigo s/n, Sisal, Hunucmá 97356, Mexico; mmm@ciencias.unam.mx (M.M.); ariadnasa@ciencias.unam.mx (A.S.); escalante.k@ciencias.unam.mx (K.E.); 2Laboratorio de Inmunología y Biología Molecular, Departamento de Recursos del Mar, Centro de Investigación y de Estudios Avanzados del Instituto Politécnico Nacional-Unidad Mérida, Puerto de Abrigo s/n, Sisal, Hunucmá 97356, Mexico; rossana.rodriguez@cinvestav.mx; 3Departamento de Biología, División de Ciencias Naturales y Exactas, Universidad de Guanajuato, Guanajuato 36000, Mexico; jp.huchin@ugto.mx; 4Unidad Académica de Sistemas Arrecifales, Instituto de Ciencias del Mar y Limnología, Universidad Nacional Autónoma de México, Puerto Morelos 77580, Mexico; briones@cmarl.unam.mx (P.B.-F.); elozano@cmarl.unam.mx (E.L.-Á.)

**Keywords:** experimental infection, immune RESPONSE, PaV1, *Panulirus argus*, phenoloxidase system, viral infection

## Abstract

**Simple Summary:**

Experimental immunological challenges are widely used to corroborate the success of breeding programs for lines resistant to specific pathogens, to test the efficiency of new vaccines, and to improve immunity of cultured animals. The validation of experimental infection protocols is complex because it requires comparison with naturally infected organisms at different stages of the infection. The present study compares the immune response of lobsters under a natural process of viral infection (PaV1), versus the defense response of experimentally infected organisms. Innate immunity for infected lobsters was measured through cellular and plasmatic components. The results indicate that the immune response of organisms naturally or experimentally infected by PaV1 was similar, and provides the bases to corroborate that the immunological challenge was not exacerbated. Appropriate infection protocols can be useful for research aimed at increasing resistance to infectious diseases and reducing the use of antibiotics in aquaculture.

**Abstract:**

Experimental infections have been used to better comprehend the immune system of organisms, and to probe for additives that generate greater resistance and help reduce antibiotic use in aquaculture. We compared the immune response of juveniles of the Caribbean spiny lobster, *Panulirus argus*, infected naturally with *Panulirus argus* virus 1 (PaV1) versus organisms infected experimentally, to determine the analogy between both infectious processes. The immunological response was measured by hemagglutination activity, hemocyte count, and total phenoloxidase activity in plasma and hemocytes in 211 individuals that were either naturally infected (110), or had been injected with viral inoculum and followed for six months (101). The samples were classified into the following four groups according to the severity of the infection: 0, uninfected; 1, lightly; 2, moderately; and 3, severely infected), which was determined on the basis of PCR and histological criteria. A permutational MANOVA showed that both the origin (natural and experimental), and the severity of the infection contributed significantly to explain the variation in the immune response of lobsters. The lack of significance of the interaction term indicated that the immunological response changed with the severity of the infection in a similar way, regardless of its origin. The results of the present study suggest that the experimental viral infection of PaV1 produces a defense response similar to the natural pathways of contagion, and provides the bases to validate an immunological challenge protocol for the first time in crustaceans. The discussion includes the perspective of the conceptual models of immune response within an ecological context.

## 1. Introduction

Numerous crustaceans, including clawed lobsters, penaeid shrimp, crabs and spiny lobsters, support valuable fisheries and aquaculture production around the world. According to the Food and Agriculture Organization of the United Nations [1], an estimated 5.7 million metric tons (live weight) of crustacean were produced in 2018. Many of these species are adversely affected by a number of diseases that significantly impact profitability [2,3]. Notable examples include the white spot syndrome virus (WSSV) of penaeid shrimp [4], *Hematodinium* and *Hematodinium*-like diseases of several crab and lobster species, and epizootic shell disease of the American lobster, *Homarus americanus* [5,6].

Considerable losses due to infectious diseases have had a devastating impact on aquaculture over the past three decades. Therefore, there is great interest in enhancing the immunity of cultivated species. For this purpose, experimental infections have been widely used to better understand the immune system of organisms, and to probe for additives that generate greater immunity and help reduce antibiotic use [7,8]. Other alternatives to mitigate the impacts of disease in aquaculture are breeding programs that include criteria to select families with a higher resistance to specific pathogens [9,10]. Shrimp breeders have focused most of their efforts on developing families of shrimp with resistance to Taura syndrome virus (TSV) and WSSV. In these programs, experimental challenges are applied to specific pathogens to determine the increase in immunity of the selected families and estimate the heritability traits.

Despite the numerous studies carried out in crustaceans, particularly *Litopenaeus vannamei* and its immune response to WSSV, there are no studies comparing the immune response of naturally infected organisms versus experimentally infected organisms. In vertebrate species (rats, dogs, birds and cats), this has been carried out to substantiate the sensitivity of diagnostic techniques and their relationship with pathological development [11,12] and also to confirm that the experimental challenge triggers a similar process of infection that occurs through natural pathways of contagion [13]. The study of natural infections and controlled challenges has also allowed for a better understanding of the interactions between the host and the pathogen. Until now, the invertebrate immune response strategy has generally been considered either resistance or death. Recent reviews have discussed the different pathways of defense as a host-regulated immune response [14]. Once infected, the following two strategies are offered to the host: parasite clearing (resistance), or withstanding the infection while paying a low fitness cost (tolerance) [15]. The plasticity of the immune response has implications for our understanding of host strategies for coping to the pathogens, and should be considered in validating experimental infection protocols.

A major obstacle facing studies that compare natural versus experimental infections is the underlying difficulty of obtaining wild organisms with different levels of a given infection. In addition, preliminary studies are required to investigate the prevalence of specific infections in wild populations. Furthermore, it is necessary to apply the appropriate diagnostic tests to determine the progress of the pathological process. While PCR is a powerful tool for pathogen detection, the detection of pathogen sequences (DNA or RNA) alone does not necessarily mean the hosts are infected with the pathogen [2]. Therefore, efforts must be made to confirm the infection and determine its progress utilizing multiple lines of evidence including in situ hybridization (ISH), histology, quantitative PCR, immunological indicators or electron microscopy (principally for viruses).

Some epidemiological studies on species such as *Nephrops norvegicus*, *Panulirus argus*, and *Callinectes sapidus* have yielded invaluable insights into how disease affects wild juveniles [3,6]. *Panulirus argus* virus 1 (PaV1) is a pathogenic virus in the Caribbean spiny lobster. It is an unenveloped, icosahedral, DNA virus recently proposed as Mininucleoviridae [16] that infects host hemocytes, spongy connective tissues, and other mesoderm derived cells [17,18]. Macroscopic signs of PaV1 infection include a milky hemolymph that does not clot, a reddish discoloration of the carapace, lethargy and, occasionally, fouling of carapace by epibionts, indicating suppression of molt [17,19]. PaV1 was first discovered in Caribbean spiny lobsters from the Florida Keys (USA) in 1999 [17], and then in Mexico [20], Belize [21] and Cuba [22]. Recent data have shown that PaV1 occurred primarily in the northern areas of the Caribbean [23].

Prevalence of PaV1 is inversely proportional to lobster size [17,18], with patent infections among juveniles reaching 50% in some areas [19]. Although disease is less evident in adults (>76 mm carapace length) (CL) [17], PaV1 prevalence of 11% with no signs of active infection has been recorded [6,23]. PaV1 has been transmitted to healthy lobsters via inoculation, prolonged contact with infected lobsters, ingestion of infected tissue, and over short distances through water [24]. The possibility of vertical transmission has been suggested by observing virus-infected hemocytes in the hemal sinuses of the ovary of female lobsters [22].

PaV1 has been implicated as a possible contributor to the decline in spiny lobster landings from the Florida Keys [17]. It is lethal to juveniles and causes fundamental changes to infected and healthy lobster ecology, which has ramifications for survival and fishery dynamics [3,5]. Several studies have examined the prevalence, pathogenicity, and population dynamics of the pathogen in the lobster host, but so far, it has not been compared with the immune responses to PaV1 in natural and experimentally infected juveniles to determine if a controlled challenge triggers a similar process of infection that occurs through natural pathway contagion. Immune response studies in natural infections of marine invertebrates are scarce. *Panulirus argus* juveniles offer the opportunity to complement laboratory studies under controlled conditions with those conducted under the natural situation. Here, we compared the immune response of juveniles under a natural process of PaV1 infection [25] versus the immune response of experimentally infected juveniles to achieve a better understanding of the defense response to PaV1 infection, and to provide information to validate an immunological challenge protocol in crustaceans.

## 2. Materials and Methods

### 2.1. Naturally Infected Juveniles

Spiny lobsters were collected from the Puerto Morelos reef lagoon, located on the Caribbean coast of Mexico (20°51′ N, 86°53′ W) by SCUBA troughs, a sampling directed for organisms that exhibit macroscopic signs of PaV1 infection (initially categorized as ‘infected’), and lobsters that do not not exhibit these signs (initially categorized as ‘healthy’). In total, 21 infected lobsters (17.9–50.3 mm CL) and 35 healthy lobsters (38.2–72.3 mm CL) were collected. Then, these were transported to the laboratory in separate containers (by presumptive infection and health condition), and placed into 2 separate holding tanks (3 m in diameter, 80 cm in depth, with ~5.7 m^3^ of seawater, salinity of 35 PSU, in a flow-through seawater system with an exchange rate of 300% per day). Before extracting hemolymph, the lobsters were placed in seawater 5 °C lower than the temperature of the water of the holding tanks to reduce their metabolic activity to minimize the handling effects on the hemolymph components [26]. Infection by PaV1 was confirmed by polymerase chain reaction (PCR) using the primer designed by Montgomery-Fullerton et al. [27]. The severity of infection was analyzed by histology of hepatopancreas, gills, muscle, midgut, and heart tissues and categorized according to Li et al. [18], and Pascual et al. [25] (Table 1). 

### 2.2. Preparation of the Viral Inoculum

Four lobsters (34.0–58.3 mm CL) with clear evidence of PaV1 infection were collected from the Puerto Morelos reef lagoon. Infection was first confirmed by PCR, and later by histology. The inoculum was prepared from the hepatopancreas of four infected organisms that were dissected; 8.87 g of tissue were macerated with 100 mL of buffered saline cold phosphate according to Manjusha et al. [28] (NaCl 8 g, KCl 0.2 g, Na_2_HPO_4_ 1.15 g, KH_2_PO_4_ 0.2 g, distilled water 1000 mL, pH 7.4; at 4 °C for 5 min). The mixture was centrifuged at 14,000× *g* for 20 min. at 4 °C. Supernatant was filtered on a 0.45 μm Millipore and stored at −20 °C. The control inoculum was prepared in the same manner as the viral inoculum, but using the hepatopancreas of one healthy adult lobster (81.3 mm LC), whose PaV1 negative condition was previously confirmed by single and double PCR [27]. The entire process of preparation and injection of the viral inoculum in lobsters was carried out under strict aseptic conditions and on the same day. The criteria for inoculum infectivity were based on the Koch’s postulate, designed to establish a causative relationship between a microbe and a disease. (1) The microorganism must be found in abundance in all organisms suffering from the disease, but should not be found in healthy organisms. (2) The microorganism must be isolated from a diseased organism. (3) The isolated microorganism should cause disease when introduced into a healthy organism. (4) The microorganism must be re-isolated from the inoculated, diseased experimental host and identified as being identical to the original specific causative agent.

### 2.3. Experimental Challenge

A total of 145 lobsters (25–79 mm LC) without clinical signs of PaV1 infection were captured and transported, as mentioned above. The lobsters were acclimated in holding tanks with a flow-through seawater system for one week prior to experiments. Only PaV1-free lobsters were used for the experiments, and 100 µL of hemolymph were obtained with a sterile syringe and used for PCR analysis to verify the absence of the virus. Then, the viral inoculum and control preparation were injected through the arthrodial membrane between the basis and ischium of the 5th pereopod. In all cases, a 70% ethanol spray was used to sterilize the area around the injection site. Each lobster was injected with 1 µL of inoculum per gram weight. To reduce stress management, the individual weight of each lobster was estimated using the equation log weight (g) = 2.4306 L; og CL (mm)–2.0048 [29].

PaV1-infected and control animals (80 and 60, respectively) were maintained in separate tanks (3 m in diameter, 80 cm in depth, with 5.6 m^3^ of seawater) that included a flow-through seawater system, with an exchange rate of 300% per day. The water was drawn directly from the sea and treated with ozone (Carbars, 48–60 g/h of ozone). The wastewater discharged from the experiment was also treated with ozone, and was injected into a well at a depth of 90 m to prevent its spread to the environment, according to Xu et al. [30]. Shelters were placed within each tank. The lobsters were fed to satiation every three days with a mixture of mussels, squid and shrimp during the acclimatization period and throughout the 187 days of the experiment. The tanks were cleaned of food debris and feces the day after feeding. To prevent viral contamination, the material and equipment used for cleaning tanks were different for each of the experimental groups. Physico-chemical water conditions were periodically monitored (pH: 7.4–8.4, salinity: 35 ± 1 psu, dissolved oxygen: >5.0 mg/L, and temperature: 24–27 °C ± 1 pH). The animals were observed daily for morbidity and mortality.

### 2.4. Sample Collection

The samples were collected at random and without replacement. Hemolymph and tissue were collected from 6 individuals per group at intervals of 15, 36, 57, 81, 105, 132, 159, and at 187 days post-injection (dpi). Before obtaining the hemolymph, the lobsters were kept in cold water (5 °C less than the temperature in the experimental system) as described previously. After swabbing the exoskeleton with 70% ethanol, ~300 μL of hemolymph was withdrawn from the base of one of the 5th pereopods using a sterile 1 mL disposable syringe; the sample was immediately placed on parafilm over a flat freeze container to avoid clotting [25]. Sub-samples were taken immediately to perform laboratory tests. Subsequently, the lobsters were weighed on a digital scale (Sartorius), sex was recorded, and the lobsters were observed for any clinical signs of disease (white hemolymph and reddish discoloration of the exoskeleton). Once the lobsters were euthanized, tissue samples were harvested (hepatopancreas, gills, muscle, midgut, and heart) for histological analysis. The tip of one pleopod was excised to score molt stage by microscopy [31].

### 2.5. Immunological Tests

Solutions were prepared using pyrogen-free water to avoid activation of the immune system by endotoxins. All glassware was previously washed with Etoxa-clean (Sigma-Aldrich, St. Louis, MO, USA).

### 2.6. Preparation of Plasma and Degranulated Hemocytes

Hemolymph was diluted (1:3) with pre-chilled (8 °C) anticoagulant according to Hernández-López et al. [32], who titrated the appropriate concentration of NaCl to avoid lysis of hemocytes in *P. argus* (350 mM NaCl, 10 mM KCl, 10 mM HEPES, 10 mM EDTA-Na_2_, pH 7.3; 850 mOsm kg^−1^). The sample was then centrifuged at 800× *g* for 5 min at 4 °C to separate the plasma, which was used to evaluate phenoloxidase activity and hemagglutination activity. The cellular pellet from each sample was washed with anticoagulant and centrifuged as described above. Then, the cellular pellet was re-suspended 30 times with cacodilic buffer (cacodilic acid 10 mM; CaCl 10 mM; pH 7.0) in equal volume (hemolymph plus anticoagulant), and centrifuged at 13,000× *g* for 5 min at 4 °C. The supernatant was used to evaluate the phenoloxidase activity from degranulated hemocytes.

### 2.7. Hemocyte Counts

A sample of 25 µL of hemolymph was diluted 1:4 with Alsever’s solution (115 mM glucose; 30 mM sodium citrate; EDTA-Na_2_ 10 mM; 338 mM NaCl) and 10% of formaldehyde (*v*/*v*) according to Le Moullac et al. [33]. The samples were kept at 2–8 °C until further analyses (7 days approximately). Hemocytes were counted in a Neubauer chamber. Samples from each lobster were analyzed in duplicate, yielding a minimum area count of 0.04 mm^3^.

### 2.8. Hemagglutination Activity

Hemagglutination activity in lobster plasma was assayed in microtiter U plates by a two-fold serial dilution; 50 μL of plasma was diluted twice in saline solution and then mixed with an equal volume of erythrocytes solution. Human blood (type O+) was obtained from a local blood bank. Before use, the erythrocytes were washed three times with 0.9% saline solution, centrifuged at 380× *g* at 25 °C for 5 min and then adjusted to a final volume of 2%. After 2 h of incubation at room temperature (26 ± 2 °C), hemagglutination was observed. The results were expressed as the inverse of the last dilution, showing visible agglutinating activity.

### 2.9. Total Phenoloxidase Activity (PO)

Total phenoloxidase activity was measured by spectrophotometry in a microplate to detect the formation of dopachrome produced [34,35]. The technique was adjusted for *P. argus* according to Pascual et al. [25]. Briefly, 100 µL of plasma or degranulated hemocytes were incubated for 10 min at RT with 100 µL of trypsin (1 mg/mL) to transform the prophenoloxidase into phenoloxidase. Then, 150 µL of from L-dihydroxyphenylalanine (L-DOPA, Sigma D9628; 3 mg/mL) was added. Absorbance was measured at 490 nm for 30 s in a microplate reader (Benchmark Plus—Biorad). The results were expressed as an increment of 0.001 in optical density.

### 2.10. PCR

DNA was extracted from hemocytes using aseptic techniques to avoid sample cross-contamination. Briefly, hemolymph stored in 1.5 mL Eppendorf tubes was thawed at ambient temperature for 15 min and centrifuged at 3000× *g* for 1 min. The pellet containing intact hemocytes, cell debris, and clotted serum proteins (~30 mg) was homogenized in 300 μL of 10% Chelex-100 (Sigma-Aldrich) containing 20 μL of 20 mg ml^−1^ proteinase K by agitation for 10 s and incubated at 56 °C for 3 h and 94 °C for 10 min. After being centrifuged at 3000× *g* for 3 min, the supernatant fluid containing DNA was carefully transferred into a sterile tube and stored at −20 °C. DNA quality and quantity were confirmed by determining the absorbance ratio OD_260_/OD_280_ using a NanoDrop 2000c spectrophotometer (Thermo-Scientific, Waltham, MA, USA), and chromosomal DNA integrity was assessed by resolving DNA in 1% agarose gels.

PaV1 DNA was amplified by PCR in a 25 μL reaction containing 1 μL extracted DNA, 0.33 μM of each primer 45aF and 543aR [28], 2.5 mM MgCl2 (Promega), 0.6× reaction buffer (Promega), 0.4 mM dNTP mixture (Promega), and 0.75 U Taq DNA polymerase (Promega). The thermal cycling conditions were 1 cycle for 94 °C for 10 min followed by 30 cycles of 94 °C for 30 s, 63 °C for 30 s, 72 °C for 1 min; followed by 72 °C for 10 min. The presence of the expected 499 bp PaV1 amplicon was determined by resolving 5 μL of the PCR product and 3 μL of loading buffer in a 2% agarose gel containing 0.1% ethidium bromide and DNA visualization using UV illumination (MiniBis Pro^®^). Ultrapure water and hemocyte DNA extracted from lobsters heavily infected with PaV1 were used as negative and positive controls, respectively [22].

### 2.11. Histology

Sample portions of hepatopancreas, gills, muscle, midgut, and heart were fixed in Davidson’s AFA fixative (alcohol, formalin, acetic acid) to determine the severity of PaV1-infection. Tissue were processed, embedded, sectioned (thickness: 5–6 µm), and stained with hematoxylin and eosin (H&E) or Pinkerton’s stain, following routine histological protocols [36,37]. Sections were analyzed to determine histopathological changes. The degree of severity was determined according to Li et al. [18] and Pascual et al. [25], from non-infected lobsters to heavily infected lobsters (grades 0–3) (Table 1).

### 2.12. Determination of Molt Stage and Size

Each lobster was measured (CL) with a Vernier caliper (±0.1 mm), immediately after hemolymph sampling. Molt stage was determined according to Lyle and MacDonald [31].

### 2.13. Statistical Analysis

Only lobsters in inter-molt were used for the statistical analyses. The immunological response of *P. argus* to infection by PaV1 (0–3 severity grades) was analyzed using non-metric multidimensional scaling (NMDS). Hemagglutination activity, hemocyte count, and phenoloxidase activity in plasma and hemocytes were evaluated in hemolymph samples taken from 211 individuals that were either naturally infected (110), or had been injected a viral inoculum 6 months earlier (101). Samples were also classified into the following four groups according to the severity of the infection: 0, uninfected; 1, lightly; 2, moderately; and 3, severely infected), which was determined on the basis of PCR and histological criteria. Dissimilarity measures between lobster samples were obtained using the reciprocal of Gower’s index (*S_17_*) [38], but the data were otherwise left untransformed. 

To compare the immunological response of lobsters from different groups, a permutational MANOVA [39] was applied on the dissimilarity matrix. The underlying experimental design was a two-way model with the origin of the infection (O) as a fixed factor with two levels (naturally and experimentally infected), and severity of the infection (S) as a fixed factor with four levels (0, 1, 2 and 3). Carapace length was used as a covariate to control for differences due to lobster size. The interaction term between origin and severity (O × S) was analyzed to explore whether changes in the severity of the infection were similar between naturally and experimentally infected lobsters. For each source of variation, 999 unrestricted permutations of raw data were used to obtain the empirical distribution of *pseudo-F* values [40]. Paired comparisons between relevant centroids were carried out following a similar procedure by calculating empirical *pseudo-t* values. The final three-dimensional configurations of the samples obtained through NMDS were represented in a two-dimensional figure to facilitate visualization and graphical interpretation of the results. The distances between centroids of severity of the infection were also visualized following this procedure. For representation purposes, box plots were additionally produced to show mean values and dispersion of each immunological descriptor both by origin and severity of the infection.

To compare the immunological response of lobsters from different groups, a permutational MANOVA [39] was applied on the dissimilarity matrix. The underlying experimental design was a two-way model with the origin of the infection (O) as a fixed factor with two levels (naturally and experimentally infected), and severity of the infection (S) as a fixed factor with four levels (0, 1, 2 and 3). Carapace length was used as a covariate to control for differences due to lobster size. The interaction term between origin and severity (O × S) was analyzed to explore whether changes in the severity of the infection were similar between naturally and experimentally infected lobsters. For each source of variation, 999 unrestricted permutations of raw data were used to obtain the empirical distribution of *pseudo-F* values [40]. Paired comparisons between relevant centroids were carried out following a similar procedure by calculating empirical *pseudo-t* values. The final three-dimensional configurations of the samples obtained through NMDS were represented in a two-dimensional figure to facilitate visualization and graphical interpretation of the results. The distances between centroids of severity of the infection were also visualized following this procedure. For representation purposes, box plots were additionally produced to show mean values and dispersion of each immunological descriptor both by origin and severity of the infection.

## 3. Results

The three-dimensional NMDS showed an effective separation of the samples according to the severity of the viral infection, with decreasing values of hemagglutination and hemocytes as the severity increased (Figure 1A), whereas phenoloxidase activity in plasma only weakly corresponded to sample ordination. By contrast, phenoloxidase activity in hemocytes increased with the severity of the infection (Figure 2). A stress value of 0.11 was obtained with the three-dimensional plot, indicating a sufficiently adequate configuration that requires caution in graphical interpretation [40].

The permutational MANOVA showed that both the origin and severity of the infection contributed significantly to explain the variation in the immune response of lobsters (*pseudo-F* = 11.95; *p* = 0.001 and *pseudo-F* = 8.94; *p* = 0.001, respectively) (Table 2). The interaction term, however, was not significant, indicating that the immunological response changed with the severity of the infection in a similar way, regardless of its origin (naturally and experimentally infected; *pseudo-F* = 0.99; *p* = 0.452) (Table 2). Pairwise comparisons between centroids showed that the response in severely infected lobsters differed from that in moderately infected ones (*pseudo-t* = 1.93; *p* < 0.05); this in turn differed from the response in lightly infected (*pseudo-t* = 2.32; *p* < 0.01); and the response in lightly infected lobsters differed from uninfected animals (*pseudo-t* = 2.53; *p* < 0.05) (Figure 1B). The analysis also showed a significant contribution of carapace length in the model (Table 2), suggesting there is a considerable variation in these immunological descriptors that can be attributed to lobster size.

## 4. Discussion

The results of the present study suggest that an experimental infection produces a similar response to that generated by a natural infection. In healthy organisms, the homeostatic mechanisms maintain a constant internal environment in the face of changing conditions. During infection, the immune response is modulated by an integrated neuro-immune network, which potentiates innate immunity, controls potential harmful effects and also addresses metabolic and nutritional modifications supporting immune function [41]. Hemocytes are the first line of defense because they participate directly in the processes of recognition, processing and amplification of the immune response [42,43]. Naturally and experimentally PaV1-infected lobsters show a substantial decrease in the total concentration of hemocytes associated with the severity of infection. This result is consistent with virus predilection for host hyalinocytes and semigranulocytes in the hemolymph [17,18], and partially explains the inability of the hemolymph to coagulate in advanced PaV1-infected lobsters because crustacean hemolymph coagulation is initiated by the release of transglutaminase from the hemocytes [44]. Aono and Mori [45] described that during the coagulation process in the spiny lobster *Panulirus japonicus* (Von Siebold, 1824), the cytolysis of hyaline and semigranular cells releases the enzyme transglutaminase. The transglutaminase levels in these cells are four times greater than in granular cells and, once released, the enzyme catalyzes the gelation of the plasma. The reduction in hyaline and semigranular cells in severely infected lobsters would reduce the activity of this enzyme that is necessary in the coagulation process. In addition to this, in shrimps, the clotting protein is synthesized in the sub-cuticular epidermis and in the heart [46,47]. In heavily infected lobsters, viral particles can be found in the spongy connective tissue surrounding several organs, including the gills, heart, hindgut, glial cells around the optic nerves, cuticular epidermis and foregut, with the hepatopancreas showing marked atrophy [17,18]. The description made in naturally infected lobsters coincides with the observations of viral inclusions in experimentally infected organisms (number of viral inclusions and damage per tissue (Table 3).

The amplification of an invertebrate’s immune response is associated with the phenoloxidase system (PO). This innate immune reaction provides toxic quinone substances and melanin that physically encapsulate pathogens, and participate in the wound healing process [48,49]. Juveniles of *P. argus* infected by PaV1 (both naturally and experimentally) showed different patterns of PO activity, depending on whether this activity was evaluated in plasma or in degranulated hemocytes. Plasma PO activity remained relatively constant with the progression of infection, whereas PO activity in hemocytes increased. This apparent disparity can be explained by the characteristics of PO molecules. Perdomo-Morales et al. [50], using electrophoretic studies (combining SDS-PAGE and native PAGE), indicated that different proteins produced the phenoloxidase activity found in the hemocyte lysate supernatant and plasma. As in most crustaceans, *P. argus* contains a prophenoloxidase (proPO) activating system in the hemocytes, whereas plasma activity appears to be produced by hemocyanin. According to Herrera-Salvatierra [51], the proPO and PO enzymes had a 1:1 ratio in uninfected lobsters, but this balance disappeared in natural PaV1-infected lobsters, which showed an increase in proPO and a decrease in PO. However, more work needs to be done to understand the genetic expression and the mechanisms of regulation of the various components of the phenoloxidase system.

Hemocyanin constitutes a high percentage of the total proteins in hemolymph. In wild lobsters heavily infected with PaV1, the concentration of total proteins in plasma decreased considerably [25], but the activity of plasma PO remains constant, and PO activity in hemocytes increases, suggesting a compensatory response of the PO system, despite the reduction in hemocytes and in plasmatic proteins. These results may reflect the multiple functionalities of hemocyanin and hemocyanin-derived peptides, which have been linked to key aspects of innate immunity, in particular antiviral and phenoloxidase-like activities [52]. According to Adachi et al. [53], this pigment can be enzymatically cleaved to produce an enzyme with similar functionality to phenoloxidase, thereby yielding an abundant pool of potential enzymes circulating within the hemolymph. 

The activation of proPO needs to be a tightly controlled process. First, the activation of PO system by microbial products denominated pathogen-associated molecular patterns (PAMPS), such as b-1,3-glucans from fungi and peptidoglycans or lipopolysaccharides from bacteria, after engaging with specific recognition proteins. In addition, enzymes and metabolites released from pathogens and tissue damage inflicted by mechanical wounding also lead to proPO activation [54]. Recognition of PAMPs is achieved via a spectrum of pattern recognition proteins (PRPs) and this interaction is instrumental in the activation of several key pathways and the release of a battery of active effector molecules [55,56]. Immulectins are members of the lectin superfamily, and they function as humoral pattern recognition receptors in innate immunity. Different lectins have been described from the hepatopancreas and/or hemocytes in penaeid shrimps [57,58], and these lectins have been shown to be transcriptionally up-regulated in response to infection with Gram-negative bacteria and some viruses, including WSSV [59]. This can be inferred by hemagglutination, which is a specific form of agglutination that involves red blood cells, and has been used to evaluate lectin recognition activity. In the present study, juveniles infected by PaV1 showed a clear decrease in hemagglutination in lobsters with an advanced degree of infection, suggesting an altered capacity of recognition as the disease progresses, potentially increasing their susceptibility to opportunistic pathogens. Pascual et al. [25], observed an increase of five times more infestation of ciliates in gills of naturally infected lobsters (61.1%) than in healthy lobsters (11.5%). Further studies are needed to examine if opportunistic pathogens increase mortality rates in PaV1-infected *P. argus*.

The observed decrease in hemocyte concentration and hemagglutination in heavily infected juveniles can reflect the damage caused by the pathogen, and it could also be the result of physiological wear due to a prolonged immune response (>80 days) [18]. Otálora et al. [60] evaluated the metabolic cost of the activation of immune response in the fish-eating Myotis (*Myotis vivesi*). They observed that an injection of lipopolysaccharide produced a significant increase in resting metabolic rate (140–185%), and substantial body mass loss. The extensive costs of pathogenesis occur because of the consequences that result from the development of disease, for example, the exploitation of host resources by the pathogen, host tissue degradation, immune suppression, or increased susceptibility to other pathogens [13,61]. The reduced activity of some immunological effectors observed in infected *P. argus* could be associated with the energy demand that is implied in a chronic disease such as PaV1. Importantly, in the wild, the criteria to detect PaV1-infected lobsters is a whitish hemolymph that does not coagulate and reddish carapace length [62]. In this study, infected lobsters were PCR-positive and had Cowdry type viral particles at 15-dpi, whereas the withe hemolymph was detected at day 81 pi. Clearly, the immunological evaluations performed here provide novel evidence of the detrimental effect of PaV1 in experimental lobsters that can be extrapolated to the effect of the virus in the wild.

A 100% of mortality was observed in juveniles of *P. argus* [17], and a significant decrease in total hemocyte count and severe alterations in some constituents in the hemolymph, such as glucose, hemocyanin, phosphorus and acylglycerides were observed [18]. On the other hand, wild lobsters exhibiting macroscopic signs of PaV1 infection had significantly lower concentrations of serum or plasmatic proteins, indicating poor nutritional condition [6,25]. Histopathological observations, used to determine the severity of the PaV1 infection, show damage and severe tissue degradation, catabolism of hepatopancreatic cells, and metabolic impairment to extract nutrients from food [17,18]. PaV1 causes profound damage to tissues and the immune response in heavily infected juveniles, revealing widespread deterioration, from which they are probably not able to recover. Based on the Matzinger model of danger/damage, Moreno-García et al. [15] proposed a quantitative model to explain the occurrence of either resistance or tolerance, and distinguish between immune strategies, including evasion and suppression of the host’s immune system by the pathogen. As invertebrates have developed strategies to fight infections, some pathogens have acquired mechanisms to evade or suppress host immune responses [63,64]. The pathogen capacity for suppression leads to unhindered growth, exceeding the damage threshold, and eventually leading to host death. However, it is unclear whether this result reflects immune suppression caused by PaV1 and/or an inability of PaV1-infected lobsters to maintain the defense response and to repair the tissues. 

In recent proteomic analyses, we analyzed molecular changes in lobsters naturally infected (PaV1) by isobaric tags for relative and absolute quantitation (iTRAQ) [37], in which a total of 636 proteins were identified, with 68 down-regulated and 71 up-regulated proteins. Among the down-regulated proteins, several enzymes involved in the metabolism of hormones, lipids and digestive enzymes were identified, while proteins associated with the histone core, protein synthesis, immune response and RNA regulation were up-regulated. Anorexia appears to be a consequence of the inability of a PaV1-infected lobster to digest its food, due to its incapability to metabolize lipids and carbohydrates. This low status associated with the deregulation of neurotransmitters and neuromodulators causes a significant reduction in the escape maneuvers. The host alterations induced by this pathogen at the molecular level suggest that the PaV1 virus might take control of the genetic machinery to promote its assembly. Transcriptomic changes in the gut of juvenile naturally infected with PaV1 were recently published [65]. Alterations in transcripts suggest several mechanisms that PaV1 employ to take control of the molecular machinery of the cell for its own benefit (e.g., transcriptional, post-translational, cytoskeletal, or transport regulation). This information reveals various mechanisms that may be related to the high pathogenicity of the virus.

The pathogenic strategy of suppression is related to high evolutionary virulence [15], which is in agreement with research on PaV1. The genetic diversity of PaV1 on a Caribbean-wide scale indicates that sequence variation in the viral DNA fragment is high, with 61 unique alleles and endemic strains identified in 9 areas [23]. In addition, PaV1 has been detected in *P. argus* postlarvae (puerulus), implying that the virus may disperse through the Caribbean within the long-lived (5 to 7 months) planktonic phyllosoma larvae, an efficient mechanism for the dispersal of viruses in the sea [66]. Mechanisms of evasion and suppression usually occur through an active process of the microorganism that hinders the detection cascades or the immune reactions of the host [15]. Further studies are needed to determine viral and host genetics/biology to improve the understanding of the prognosis of PaV1 infection in different ontogenetic faces.

The study of diseases within an ecological context requires the consideration of an overall consensus in favor of the most-used immune response effectors. Some defense mechanisms have been selected, such as the critical response pathways common to all phyla of invertebrates; among them, the phenoloxidase pathway, peripheral phagocytic cells, the cytotoxic effector responses and the antimicrobial compounds have been renowned [67]. In the present work, the phenoloxidase system was evaluated through the total activity in the different compartments (cellular and humoral), and included the activity of plasma recognition proteins through lectin activity, evaluated by hemagglutination. Phenoloxidase activity observed in lobsters infected by PaV1 may partially explain the long period of the PaV1 infectious process, suggesting that the survival of some lobsters may be due to an adaptive response. The immunological response that attempts to overcome chronical viral infection has already been observed in *Macrobrachium rosembergii* infected with WSSV [68]. Immunological parameters, such as proPO, and clotting time, in WSSV-injected *M. rosenbergii* were found to be significantly higher than those of the control groups, whereas hemocytes concentration and superoxide dismutase activity were significantly lower when compared to control groups.

In the present study, the origin of the infection (natural and experimental) contributed significantly to explain the variation in the immune response of lobsters. This could be related to the viral DNA concentration in the inoculum. For example, a similar study on dogs, either naturally or experimentally infected with *Ehrlichia canis,* indicated that the response to inoculation was dose dependent; with natural infection being associated with lower numbers of DNA copies than the numbers found in experimentally infected dogs [11]. The studies of immune response in natural infections of marine shellfish are scarce. In addition to the difficulty of obtaining wild organisms with different levels of the studied infection, a background on the prevalence and information on population dynamics is required. Due to these difficulties, experimental infections have been widely used, but it raises the following question: how similar is the host–pathogen interaction in an experimental versus a natural infection? Therefore, how appropriate is it to interpret what could happen at the population level (wild or captive) without clarifying this? The results of this study indicate that the immune response of juveniles that have been naturally or experimentally infected by PaV1 is similar, and provides the bases to validate an immunological challenge protocol for the first time in crustaceans. Experimental infections have been fundamental to better understand diseases and have great potential to develop the knowledge of environmental effects (changes in environmental temperature patrons, contamination, acidification, etc.), on host–pathogen interactions within an ecological context.

## 5. Conclusions

Innate immunity for infected lobsters was followed by the phenoloxidase system (cellular and plasmatic components), activity of plasmatic proteins of recognition (lectins), and hemocyte concentration. The multivariate analysis of the immunological evaluations allowed us to trail the immune response variation by comparing four degrees of severity of the infection by PaV1, from non-infected lobsters to heavily infected lobsters, determined on the basis of PCR and histological criteria. The permutational MANOVA showed that both the origin (naturally and experimentally infected) and severity of infection contributed significantly to the variation in the immune response of lobsters. The lack of significance of the interaction term indicated that the immunological response changed with the severity of the infection in a similar way regardless of its origin, and provides the bases to corroborate the immunological challenge protocol.

## Figures and Tables

**Figure 1 animals-12-01951-f001:**
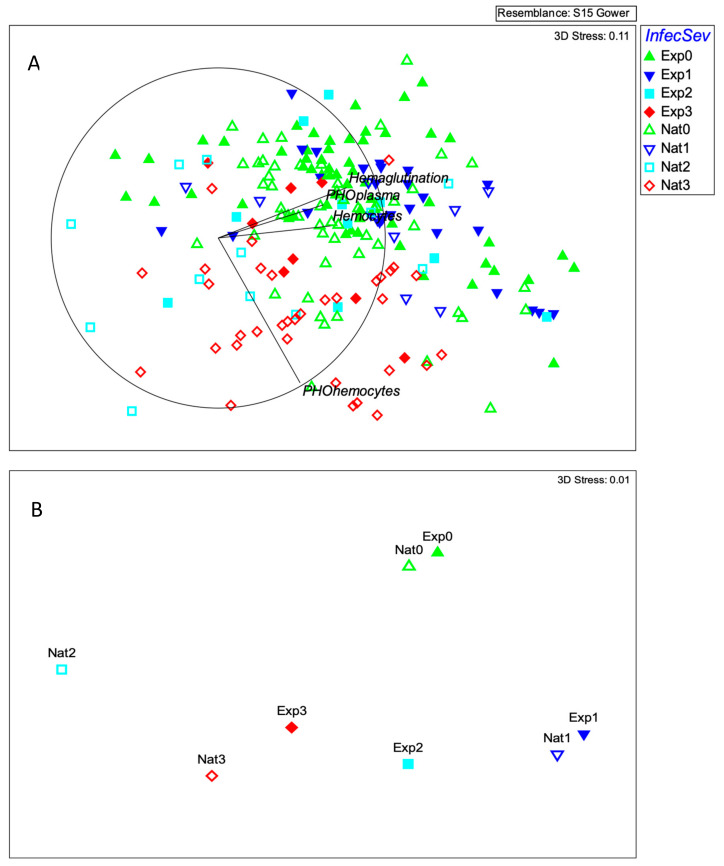
Non-metric multidimensional scaling (NMDS) of immunological response of *P. argus* to infection by PaV1; PO: total phenoloxidase activity in plasma and hemocytes (optical density, 490 nm), hemocytes concentration (cells mm^−3^), and hemagglutination (titer). The colors correspond to the severity of the infection (0: uninfected; 1: lightly; 2: moderately; and 3: severely infected, which was determined on the basis of PCR and histological criteria). Full symbols correspond to experimental challenge and empty to natural infection process; 112 organisms (**A**), and centroids of severity and origin of infection (**B**).

**Figure 2 animals-12-01951-f002:**
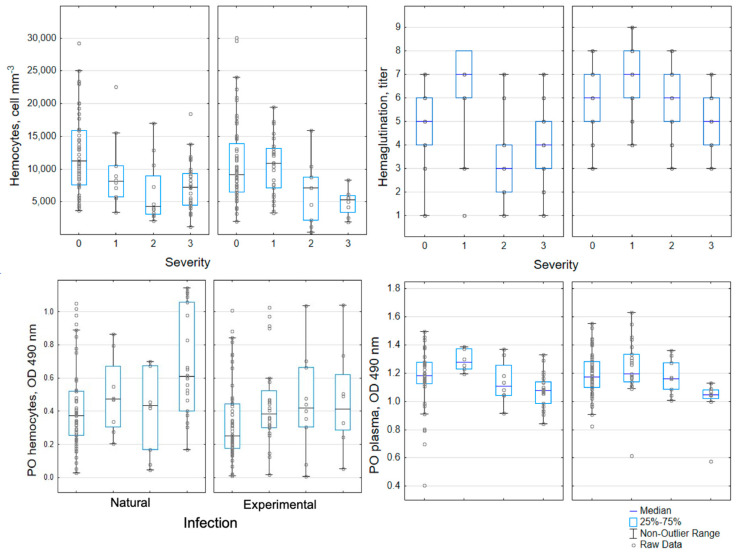
Box plots of immunological indicators measured in *Panulirus argus* either naturally or experimentally infected with PaV1. Severity of the infection (0: uninfected; 1: lightly; 2: moderately; and 3: severely infected), on the basis of PCR and histological criteria. PO: total phenoloxidase activity. Interpretation of univariate statistical features depicted in these figures must be carried out with caution, since they assume that all the response variables are uncorrelated and independent of each other.

**Table 1 animals-12-01951-t001:** Grades of severity of *Panulirus argus* virus 1 (PaV1) infection in *Panulirus argus* lobsters according to Li et al. [17] and Pascual et al. [25]. Number of organisms in each degree of severity by infection origin: natural (Nat.), and experimental (Exp.).

Infection Severity	PCR Result	Histopathological Observations	Infection Origin	
			Nat.	Exp.
**Grade 0 ** Healthy Lobsters without infection Control treatment of experimental infection	Negative	No aberrant cells with hypertrophied nuclei, no peripheral chromatin or eosinophilic inclusions Hepatopancreas and other tissues appear normal	54	57
**Grades 1** Lightly infected	Positive	Few or no infected cells present in hepatopancreas or other organs (1–10 per section)	10	26
**Grades 2** Moderately infected	Positive	Fixed phagocytes activated or infected Moderate hemocytic infiltrate Moderate obstruction of hemal sinuses Some infected cells present in organs (1–100 per section)	12	10
**Grade 3** Heavily infected	Positive	Hepatopancreatic tubules atrophied Presence of nodulations or granulomas Many infected cells present in spongy connective tissue around organs (>100 per section)	34	8

**Table 2 animals-12-01951-t002:** MANOVA with permutations applied to the values of the immunological response measurements in *Panulirus argus,* either naturally or experimentally infected with PaV1. Experimental design was a two-way model with the origin of the infection (O) as a fixed factor with two levels (naturally and experimentally infected); severity of the infection (S) as a fixed factor with four levels (0, 1, 2 and 3); and carapace length (CL) as a covariate.

Source	fd	SS	MS	*Pseudo-F*	*P*	Unique Permutations
Carapace length	1	4944.9	4944.9	21.89	0.001	999
Origin (O)	1	2703.5	2703.5	11.95	0.001	999
Severity (S)	3	6067.4	2022.5	8.93	0.001	999
O × S	3	673.9	224.63	0.99	0.452	999
Residuals	202	45720	226.34			
Total	210	6011				

**Table 3 animals-12-01951-t003:** Histological evaluation of juveniles of *Panulirus argus* experimentally infected with PaV1. d.p.i. = days post-infection. PCR+. Grades of infection 1–3 based on the scale of Li et al., 2008 and Pascual Jimenez et al., 2012.

d.p.i.	Hepatopancreas	Gills	Intestine	Heart	Muscle
15	1	1	1		
36	2	1	1	1	
57	2	1	1	2	
81	3	2	2	2	
105	3	2	2	3	1
132	3	3	2	3	1
159	3	3	2	3	1
187	3	3	2	3	2

## Data Availability

The datasets generated for this study are available upon request to the corresponding author.

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
