# Peer review of "Immune Response to Natural and Experimental Infection of *Panulirus argus* Virus 1 (PaV1) in Juveniles of Caribbean Spiny Lobster"

_animals, 2022, doi:10.3390/ani12151951_

Round 1
Reviewer 1 Report
The authors investigated the immune response to natural and experimental infection of Panulirus argus Virus 1 (PaV1) in juveniles of Caribbean spiny lobster. This manuscript (MS) was clearly written and easy to understand. This work can help the sustainability of this species farming as few studies in this area have been done. However, some minor issues significantly compromised the quality of this MS.
Minor comments
Abstract
- Line 19-21, please revise it is not clear.
- Line 28, please revise it.
- Line 32 and elsewhere, please first mention the common name plus the scientific name, and for the rest of the MS, only report the common name.
- Please reorder the keywords alphabetically and capitalize each word.
Introduction:
- Well-developed introduction and included a clear fellow and relevant points.
- Line 56, please only report crustacean annual production and not fish. It is available in FAO 2020.
- Line 57, please revise.
- Line 63, please revise “Substantial losses”.
- Line 78-80, please do not start sentences with “And”
- Line 121, please be consistent with the common name (scientific name) for the first time in the MS and then, only the common name.
- Please summarise the introduction (delete like 5-7 lines).
- Please mention the novelty of your work in the last paragraph of the introduction.
Material and methods
- Well-organized section. Clear fellow and all required details were provided.
- Please mention how many percentages of water were exchanged each day if you have monitored.
- Please check the space between words and delete extra spaces.
- Line 168, please make sure you defined each abbreviation for the first time in the MS.
- It is not clear to me why did you use MANOVA for this study. You could use the simple two-way ANOVA and the interpretation of the results was simpler.
- For each analysis, please clarify how many lobsters were taken.
Results
- Well-written section, all necessary things have been covered.
- Tables, please be consistent with the font of the text
- Please improve the quality of figure 2 and in the footnote please mention how many samples were taken for these analyses.
Discussion
- Put the subheading for the discussion section like results. Also, keep a sequence in subheading for investigated factors, in material, and method, result, and discussion. It will help readers to understand your paper easier.
- Line 410, please revise this sentence and “gene activation”.
- Line 482, please suggest this point in conclusion for future research.
- Line 524-531, I am not sure that it is a good way to compare your results with the dog. The mechanism of action is completely different. I suggest using aquatic species.
- Generally, this section was presented well, and I do not have any comments. However, please read the MS to fix some language errors
Best regards
Author Response
Response to Reviewer 1 Comments
Abstract
- Line 19-21, please revise it is not clear. Revised
- Line 28, please revise it. Revised
- Line 32 and elsewhere, please first mention the common name plus the scientific name, and for the rest of the MS, only report the common name.
- Please reorder the keywords alphabetically and capitalize each word. Done
Introduction:
- Well-developed introduction and included a clear fellow and relevant points.
- Line 56, please only report crustacean annual production and not fish. It is available in FAO 2020. Done
- Line 57, please revise. Done
- Line 63, please revise “Substantial losses”. Done
- Line 78-80, please do not start sentences with “And” Done
- Line 121, please be consistent with the common name (scientific name) for the first time in the MS and then, only the common name. Done
- Please summarise the introduction (delete like 5-7 lines). Done
- Please mention the novelty of your work in the last paragraph of the introduction. Done
Material and methods
- Well-organized section. Clear fellow and all required details were provided.
- Please mention how many percentages of water were exchanged each day if you have monitored. Information included (300 %/day)
- Please check the space between words and delete extra spaces.
- Line 168, please make sure you defined each abbreviation for the first time in the MS. Done
- It is not clear to me why did you use MANOVA for this study. You could use the simple two-way ANOVA and the interpretation of the results was simpler.
A multivariate approach was used analyse the immune response of lobsters to infection by PaV1 based on two statistical criteria: 1) The four variables evaluated (hemagglutination activity, hemocyte count, and phenoloxidase activity in plasma and hemocytes) were measured in the same experimental units, hence are correlated to each other in different degrees. This results in a high dimensional data set with much biological information residing precisely in the covariation amongst variables, information that would have gone unnoticed had we used four univariate ANOVA’s separately applied to each descriptor. 2) A permutational MANOVA was chosen as a procedure to test hypothesis regarding the effect of both factors and the interaction term specified in the underlying statistical model. This procedure has been recommended as an alternative to simple ANOVA when the residuals of the statistical model are not normally distributed, which was the case in at least two of the four response variables used to describe the data.
By using a multivariate approach, we enhanced the probability of finding differences between experimental groups if they exist. Despite the comparatively les simplicity of multivariate methods, in this case we were able to describe the immune response in a much more complete and realistic manner.
- For each analysis, please clarify how many lobsters were taken.
We are not clear what Reviewer 1 refers to with “each analysis”, since we only performed one nMDS with a permutational MANOVA that included 2 factors, 1 interaction term and 1 covariate. However, we have added the sample sizes of all 8 experimental groups in table 1:
Exp0: n = 57
Exp1: n = 26
Exp2: n = 10
Exp3: n = 8
Total: 101
Nat0: n = 54
Nat1: n = 10
Nat2: n = 12
Nat3: n = 34
Total: 110
Results
- Tables, please be consistent with the font of the text. Done
- Please improve the quality of figure 2 and in the footnote please mention how many samples were taken for these analyses.
- Figure 2 was enhanced. The number of organisms analyzed was included in Table1
Discussion
- Put the subheading for the discussion section like results. Also, keep a sequence in subheading for investigated factors, in material, and method, result, and discussion. It will help readers to understand your paper easier.
The results and discussion sections do not have subheadings because the multivariate analysis leads to a comprehensive interpretation (fig 1). Figure 2 (box plots) were additionally produced to show mean values and dispersion of each immunological descriptor both by origin and severity of the infection.
- Line 410, please revise this sentence and “gene activation”.
Thank you for the observation. It was changed by genetic expression.
- Line 482, please suggest this point in conclusion for future research.
Thank you. A recent publication (2022) on transcriptomic analysis was included in the text.
- Line 524-531, I am not sure that it is a good way to compare your results with the dog. The mechanism of action is completely different. I suggest using aquatic species.
We totally agree, however, there are really few comparative studies of the immune response to wild versus experimental infections. In the searches carried out we have only found the following works, which were mentioned in the writing:
- [11] Baneth, G.; Harrus, S.; Ohnona, F.S.; Schlesinger, Y. Longitudinal quantification of Ehrlichia canis in experimental infection with comparison to natural infection. Microbiol. 2009, 136, 321–325; doi:10.1016/j.vetmic.2008.11.022.
- [12] Atkins, C.E.; Vaden, S.L.; Arther, R.G.; Ciszewski, D.K.; Davis, W.L.; Ensley, S.M.; Chopade, N.H. Renal effects of Dirofilaria immitis in experimentally and naturally infected cats. Parasitol. 2011, 176, 317–323; doi:10.1016/j.vetpar.2011.01.016.
- [13] Bonneaud, C., Balenger, S.L.; Hill, G.E.; Russell, A.F. Experimental evidence for distinct costs of pathogenesis and immunity against a natural pathogen in a wild bird. Ecol. 2012, 21, 4787–4796; doi:10.1111/j.1365-294x.2012.05736.x.
- [14] Sorci, G. Immunity, resistance and tolerance in bird-parasite interactions, Parasite Immunol. 2013, 35; doi:10.1111/pim.12047.
Thank you very much for your comments, they have allowed us to improve the document.
Kind regards,
Authors
Reviewer 2 Report
Summary:
Pascual et al. performed comparative study of immunological responses of Panulirus argus following natural and experimental infection with P. argus Virus 1. This is to provide a basis for controlled infection protocols in this crustacean and valuable metric for investigating immune improvement interventions and corroborating the success programs that aims to breed for resistance in the cultured animals.
In my opinion, the study is well-designed and well-written and conclusions reflect the data presented. However, the authors have not sufficiently addressed an important component for ensuring reproducible controlled infection protocols – the inoculum dosage.
Immunological response to experimental infection can be dose-dependent as the authors have alluded to in the discussion of the results. However they did not address this in the design of this study, at least not in the manuscript in its current form. The inoculum used appears to be arbitrarily determined. I believe that answers to the following questions could add some standardization to the preparation and use of inoculum:
Source of inoculum: what was the severity of infection in the lobsters used?
Weight of hepatopancreas macerated: was 8.87 g the total tissue obtained from all four lobsters? Or, was this some empirically-determined value?
While the volume of inoculum per lobster weight is given, how many copies of PaV1 would this translate to per gram weight? This can be determined by a quantitative PCR.
Secondly, summary information on the 110 naturally infected lobsters is not included. How many belonged to the respective severity categories? While this is apparently integrated into the analyses as illustrated in Figures 1 & 2, a summary statistics, similar to Table 3, would be useful to the reader.
Finally, the manuscript would benefit from correction of minor typographical errors sparingly in the text e.g. lines 484, 486-487 etc.
See additional specific comments below for details of requested clarifications and corrections.
Comments:
Line 177: ‘estimation of lobster weight’ – The reference (Xu et al.) appears incorrect. Include appropriate reference for this method. How accurate is this?
Line 221: ‘several times’ – Please, indicate the exact number of times re-suspension was performed or an accurate range.
Line 288: Why were only the inter-molt stage used? What is the significance to the analysis? This should be stated.
Line 293: Number of animals used could be better explained for clarity. In line 178, it says 80 PaV1-infected but here it says 101. Could you reconcile the discrepancy?
Author Response
Response to Reviewer 2 Comments
Source of inoculum: what was the severity of infection in the lobsters used?
The hepatopancreas used to prepare the inoculum were obtained from four lobsters with clinical signs associated with very advanced stages of PaV1 disease. Animals with such clear clinical signs indicate a severe PaV1 infection (Candia-Zulbarán et al., 2012). In the article we mention that histological analysis was used to confirm the degree of severe infection, following the severity scale according to Li et al., 2008 and Pascual et al., 2012.
Candia-Zulbarán, R.; Briones-Fourzán, P.; Negrete-Soto, F.; Barradas-Ortiz, C.; Lozano-Álvarez, E. Variability in clinical prevalence of PaV1 in Caribbean spiny lobsters occupying commercial casitas over a large bay in Mexico. Dis. Aquat. Organ. 2012, 100, 125–133; doi: 10.3354/dao02452.
Weight of hepatopancreas macerated: was 8.87 g the total tissue obtained from all four lobsters?
Correct, the 8.87 g were obtained from four lobsters with severe infection, in order to have enough inoculum for all experimentally infected organisms (80), and in order to have a viral source with some biological diversity, that is, result of the host-pathogen interaction of various organisms. We rewrite the paragraph to improve the understanding of the procedure performed.
Or, was this some empirically-determined value?
The criteria for inoculum infectivity were based on the Koch´s postulate designed to establish a causative relationship between a microbe and a disease. Briefly; 1). The microorganism must be found in abundance in all organisms suffering from the disease but should not be found in healthy organisms. 2). The microorganism must be isolated from a diseased organism. 3). The isolated microorganism should cause disease when introduced into a healthy organism. 4). The microorganism must be re-isolated from the inoculated, diseased experimental host and identified as being identical to the original specific causative agent.
Effectively, we start from an empirical value. Our goal was to ensure that the animals were infected. We relied on widely used shrimp protocols for experimental infections with white spot virus, which was prepared according to Manjusha, el al., 2009 (doi:10.1016/j.jip.2009.08.011).
While the volume of inoculum per lobster weight is given, how many copies of PaV1 would this translate to per gram weight? This can be determined by a quantitative PCR.
At the time of the experiment we did not quantify the number of copies. It would have been the ideal, but we didn't have the oligos and the conditions for this type of study. The first study to quantify PaV1 viral load was standardized and published by Clark et al., 2018. https://doi.org/10.3354/dao03242
We have recently performed the standardization of PaV1 diagnosis by qPCR (unpublished data). Analysis of severely infected hepatopancreas from lobsters from the same geographical area was quantified at 7.4 x105 copies of PaV1 per µg of DNA. As the viral load was not evaluated by qPCR during the infection experiment (this study), we considered it appropriate not to include it.
Secondly, summary information on the 110 naturally infected lobsters is not included. How many belonged to the respective severity categories?
We have included this information in Table 1.
While this is apparently integrated into the analyses as illustrated in Figures 1 & 2, a summary statistics, similar to Table 3, would be useful to the reader.
We understand that Reviewer 2 suggests we include a Table with the summary statistics of all four descriptors in each of the 8 combinations of origin and severity of PaV1 infection. If we are correct in our understanding that is exactly the graphic information of Figure 2. Since these variables rarely showed symmetric distributions, we considered using the median as a measure of central tendency and the inter-quartile range as a measure of dispersion. We believe that a table with the same values would be redundant, and are confident that our graphical selection best depicts the scale and magnitudes of the response variables.
Finally, the manuscript would benefit from correction of minor typographical errors sparingly in the text e.g. lines 484, 486-487 etc. Done
Line 177: ‘estimation of lobster weight’ – The reference (Xu et al.) appears incorrect. Include appropriate reference for this method.
Thanks for the observation. The correct citation was already included:
[29] Lozano-Álvarez, E. Pesquería, Dinámica Poblacional y Manejo de la Langosta Panulirus argus (Latreille, 1804) en la Bahía de la Ascensión, Q.R., México. Ph.D. Thesis, Universidad Nacional Autónoma de México, Mexico City, Mexico, 1992.
How accurate is this?
Morphometric relationships allow testing the relationship between two variables and predicting the value of one (Y) from the other (X), whereas allometry is useful to examine how one variable scales with another (Warton et al., 2006 https://doi.org/10.1017/S1464793106007007).
The correlation between wet weight and CL has been evaluated in various ecological studies of spiny lobsters. The correlation coefficient (R2) has been indicated between 84 and 98% (p < 0.05), for example:
Pérez-González, Raúl. (2011). Catch composition of the spiny lobster Panulirus gracilis (Decapoda: Palinuridae) off the western coast of Mexico. Latin american journal of aquatic research, 39(2), 225-235. DOI: 10.3856/vol39-issue2-fulltext-4
Martínez-Calderón R, Lozano-Álvarez E, Briones-Fourzán P. 2018. Morphometric relationships and seasonal variation in size, weight, and a condition index of post-settlement stages of the Caribbean spiny lobster. PeerJ 6:e5297 https://doi.org/10.7717/peerj.5297
Line 221: ‘several times’ – Please, indicate the exact number of times re-suspension was performed or an accurate range.
Quantity was included: about 30 times.
Line 288: Why were only the inter-molt stage used? What is the significance to the analysis? This should be stated.
The ecdysis process affects the metabolism and the immune system of crustaceans, for which it is recommended to consider the molting stage in physiological and immunological analyses, with the intermolting stage being the longest (Le Mollac G. et al 1997)
Line 293: Number of animals used could be better explained for clarity. In line 178, it says 80 PaV1-infected but here it says 101. Could you reconcile the discrepancy?
The number 101 is the total of organisms used in the experimental challenge, it includes the control treatment organisms (category 0: uninfected), plus the infected ones (categories 1-3).
Thank you very much for your comments, they have allowed us to improve the document.
Kind regards,
Authors
Reviewer 3 Report
The present study compares the immune response of lobsters under a natural process of viral infection (PaV1), versus defense response of experimentally infected organisms. I have several comments and suggestions. First of all, the project idea is valuable to the research area so I am glad to see a study of this nature.
I found myself a little confused at times with regard to the wording; some text editing is necessary
Line 27 -29: not sure your intent of this statement – are you suggesting using controlled infection to reduce susceptibility to infection in aquaculture?
-Line 171: sterile gauge?? What was the gauge?
I found myself a little confused with the experimental setup and data analysis. I assume the animals with infection level of 0 were your controls, but this was never clarified. Also, for experimental animals with infection level of 0 – were these injected with any solution to control for handling and injection protocols in experimentally treated animals? These should be sham-injected with the same buffer used to inject the virus. A direct comparison of infected to uninfected should be included, but I didn’t see this, only comparison between experimentally infected and naturally infected.
Also, it looks as if animals were sampled over several time points, but why is this data never shown? Were the results different at different each time period? At what time period are the data in Figure 2 representing? Is Figure 1 all the data from each time period? A table with final numbers of animals analyzed in each group for each time period and each assay would be helpful. For example in Figure 2 for hemagglutination, is seems as if only 1 animal for Infection Level 0 for experimental?
What primers were used for PCR?
Much of your data is based on sorting animals based on infection level determined from histology, yet there are no histology images shown. It would be helpful to include some histology images of the different levels of severity used to sort animals.
Line 22: please check the spelling of cacodylate acid as it is spelled two different ways in this sentence
Line 229: how much time of storage before analysis
Line 230: was any estimate of viability conducted? Was there any dilution prior to counting?
In the discussion, your text seems to include a lot of discussion on lobster, but then I realized maybe the discussion includes use of shrimp and lobster interchangeably without clarifying which one is actually being discussed. This should be rewritten to clarify which text refers to lobster and which to shrimp. Since the paper is on shrimp, I would think the majority of references would be to shrimp.
Author Response
Response to Reviewer 3 Comments
The present study compares the immune response of lobsters under a natural process of viral infection (PaV1), versus defense response of experimentally infected organisms. I have several comments and suggestions. First of all, the project idea is valuable to the research area so I am glad to see a study of this nature.
Thank you very much for your comment, we are glad to know that you like it
I found myself a little confused at times with regard to the wording; some text editing is necessary
Line 27 -29: not sure your intent of this statement – are you suggesting using controlled infection to reduce susceptibility to infection in aquaculture?
We rewrite the sentence. We mean that experimental infections are necessary to test the efficiency of immune-additives, and to corroborate resistance to specific pathogens in genetic programs to generate lines less susceptible to infectious diseases. It is these alternatives that can help reduce the use of antibiotics in aquaculture.
Line 171: sterile gauge?? What was the gauge?
Thank you for your observation. One mistake, we mean syringe.
I found myself a little confused with the experimental setup and data analysis. I assume the animals with infection level of 0 were your controls, but this was never clarified. Also, for experimental animals with infection level of 0 – were these injected with any solution to control for handling and injection protocols in experimentally treated animals?
Indeed, the control treatment organisms were injected with hepatopancreas from a lobster without signs of the disease and free of the virus (negative PCR). Control inoculum was prepared in the same way as the viral inoculum, in order to generate the same immunological activation that could be triggered by the tissue components (not including PaV1 viral material).
This information is found in the inoculum preparation paragraph: “Control inoculum was prepared in the same manner as viral inoculum, but using the hepatopancreas of one healthy adult lobster (81.3 mm LC), whose PaV1 negative condition was previously confirmed by single and double PCR [27]. The whole process of preparation and injection of viral inoculum lobsters was performed under strict aseptic conditions”.
These should be sham-injected with the same buffer used to inject the virus. A direct comparison of infected to uninfected should be included, but I didn’t see this, only comparison between experimentally infected and naturally infected.
The comprehensive analysis includes the uninfected in category 0 infection. In the case of wild organisms, they are those animals without signs of infection, and negative to PCR test and histological analysis. In the experimental infection group, the infection category 0, included to the organisms that was negative to PaV1 (PCR evaluated in hemolymph), and then injected with the "control inoculum" (hepatopancreas of a healthy lobster - negative for PaV1 by PCR and histology).
Also, it looks as if animals were sampled over several time points, but why is this data never shown? Were the results different at different each time period?
In experimentally infected animals we analyzed by time of infection (data not shown), but the patterns of the immune response were not as clear. The immune response was much better observed when monitoring the degree of infection, since the development of the infection does not occur exactly at the same time (days post infection), since it depends on the host-pathogen interaction. In addition to this, in the case of wild lobsters infected by natural routes of infection, we do not know the post-infection time. Analyzing them by infection category allowed us to compare the defense response of both groups.
At what time period are the data in Figure 2 representing?
Incluye todas las langostas evaluadas en los 8 muestreos del experimento de infección (15, 36, 57, 81, 105, 132, 159, and at 187 dpi). También incluye a todas las langotas infectadas naturalmente, ambos grupos categorizados en los 4 grados dinfección ((0: uninfected; 1: lightly; 2: moderately; and 3: severely infected).
Is Figure 1 all the data from each time period?
Includes all lobsters evaluated in the 8 samples of the infection experiment (15, 36, 57, 81, 105, 132, 159, and at 187 dpi). It also includes all naturally infected lobsters, both groups categorized into 4 degrees of infection (0: uninfected; 1: lightly; 2: moderately; and 3: severely infected).
A table with final numbers of animals analyzed in each group for each time period and each assay would be helpful.
The number of organisms per infection category in each infection group (natural and experimental) was included in table 1:
Exp0: n = 57
Exp1: n = 26
Exp2: n = 10
Exp3: n = 8
Total: 101
Nat0: n = 54
Nat1: n = 10
Nat2: n = 12
Nat3: n = 34
Total: 110
For example in Figure 2 for hemagglutination, is seems as if only 1 animal for Infection Level 0 for experimental?
Figure 2 was improved.
What primers were used for PCR?
The presence of PaV1 was confirmed by polymerase chain reaction (PCR) using the primer designed by Montgomery-Fullerton et al. [27].
Much of your data is based on sorting animals based on infection level determined from histology, yet there are no histology images shown.
The histological description of PaV1 disease has been published in several papers (Shield et al. 2004; Li et al., 2008). In this study, histological analysis was used to confirm the severity of the infection, therefore it was not included.
It would be helpful to include some histology images of the different levels of severity used to sort animals.
We consider it appropriate to only include the description of the histological criteria used to define the degree of severity, which are found in Table 1.
Line 22: Please check the spelling of cacodylate acid as it is spelled two different ways in this sentence. Done
Line 229: how much time of storage before analysis.
Information included: 7 days approximately
Line 230: was any estimate of viability conducted?
We corroborated the effect of refrigeration time (4ºC) on the total hemocyte count for 20 days in Alsever-formaldehyde (10% v/v). There is no difference with the readings made on the same day of the hemolymph extraction. We have only performed the trypan blue test in the first 10 days (data not shown).
Was there any dilution prior to counting?
Thanks for your observation. The initial dilution was 1:4 in Alsever solution (corrected in the text). Most of the time a second dilution is not necessary.
In the discussion, your text seems to include a lot of discussion on lobster, but then I realized maybe the discussion includes use of shrimp and lobster interchangeably without clarifying which one is actually being discussed. This should be rewritten to clarify which text refers to lobster and which to shrimp. Since the paper is on shrimp, I would think the majority of references would be to shrimp.
Indeed, most of the studies on the immune response to experimental infections in marine crustaceans have been described mainly in shrimp. Studies with other decapods were also included. The organisms corresponding to each study were included in the discussion.
Thank you very much for your comments, they have allowed us to improve the document.
Kind regards,
Authors